# Development of Geospatial Data Acquisition, Modeling, and Service Technology for Digital Twin Implementation of Underground Utility Tunnel

Jaewook Lee [1], Yonghwan Lee [2] and Changhee Hong [1,*]

1 Department of Future & Smart Construction Research, Korea Institute of Civil Engineering and Building Technology, Goyang-si 10223, Republic of Korea
2 CEO, IANSIT, Incheon 21999, Republic of Korea
* Correspondence: chhong@kict.re.kr; Tel.: +82-31-910-0706

**Abstract:** In the maintenance domain of the construction industry, digital twins have been actively introduced based on the technologies of the Fourth Industrial Revolution with growing interest in three-dimensional spatial information facility management and disaster response service using digital twin technology. In particular, disasters or abnormal situations in an underground utility tunnel (UTU) can cause serious casualties and property damage since key elements of a city such as power, communications, water supply, and heating facilities are collectively accommodated in a certain underground space. This study established the methodology for implementing an underground utility tunnel through a digital twin. The novelty of this study is demonstrating a sequential procedure of implementing digital twin technology by configuring major layers such as data acquisition, modeling, and service. This methodology can serve as reference material or a training aid for implementing a digital twin of underground facilities or underground utility tunnels in the future; the proposed methodology will be verified when an actual digital twin service is provided.

**Keywords:** digital twin; underground utility tunnel; building information modeling (BIM); spatial information; modeling

## 1. Introduction

The global construction industry is pursuing the conversion to a digitalized smart construction system through the convergence of Fourth Industrial Revolution technology, spurring fierce competition to improve productivity and lead the market in the construction industry [1]. Major issues in conventional construction methods are inadequate response measures for productivity, digitalization, and aging, in addition to the need for changes in conventional construction work methods due to an expansion of contactless types of work [2,3]. In particular, the digitalization index of the construction industry is lower than that of other industries such as agriculture, manufacturing, and information communications [4]. Furthermore, the average age of infrastructure technicians is gradually increasing, while the working age population is predicted to decrease [5]. Due to the expansion of contactless types of work during the COVID-19 pandemic, changes in the construction industry are occurring through the utilization of the metaverse where virtual and real technologies involving extended reality (XR), artificial intelligence (AI), the Internet of Things (IoT), and blockchain are being applied [6]. Especially in the maintenance domain of the construction industry, digital twins have been actively introduced based on the technologies of the Fourth Industrial Revolution, as explained above.

The emergence of a digital twin is attributable to advances in various information technologies [7–9]. Various technologies for expressing the real world in three dimensions such as computer-aided design (CAD) and building information modeling (BIM) have matured,

thus enabling the design of a newly created real world (reality) in three dimensions. Other technologies such as light detection and ranging (LiDAR), photogrammetry, and Pictometry realize real objects (buildings, roads, rivers, forests, etc.) as three-dimensional spatial information, and the advancements of sensors help to identify real-world situations [10]. In addition, communications and internet networks such as 5G transmit sensed data in real time. Similarly, the circumstances for forming a digital twin have been established through the convergence of a variety of technologies including the visualization of three-dimensional models, advancements in big data and artificial intelligence (AI) technology, and simulation technologies such as agent-based models.

Owing to a growing interest in 3D spatial-information-based facility management through digital twin technology and the digitalization of deteriorated social overhead capital (SOC) and disaster response services in recent years, the utilization of various three-dimensional spatial models has also increased. Such virtual models are being used as base models in diverse areas such as facility management, urban management, and disaster management, and there is a growing need to establish and manage interior as well as exterior space information. Existing spatial information and application services have been developed mostly for exterior spaces, and data models have also been defined and developed with a focus on exterior spaces. Moreover, the models have been developed mainly to express objects and thus have insufficient property information, which limits their performance in spatial analysis. Thus, an interior space data model is needed to support the services using 3D interior space data and geographic information system (GIS) technology [11]. These data models must fundamentally provide visual information about an interior space and be capable of performing spatial analysis.

An underground utility tunnel is an urban infrastructure that collectively accommodates and manages the lifelines of a city such as power, communication, water supply, and heating facilities in a certain underground space; accordingly, the presence of an underground utility tunnel helps to avoid repeated excavation and imprudent use of underground spaces for the maintenance of underground facilities, while improving the urban landscape, preventing disasters, preserving road structures, and securing a smooth traffic flow [12]. Since an underground utility tunnel encompasses the lifelines directly associated with the life of citizens, serious chaos can occur in an underground utility tunnel as well as in ground-level spaces if a disaster happens in the tunnel.

Therefore, for the smart management of deteriorated underground utility tunnels, research must be conducted on building and linking data and modeling based on a digital twin, which is the technology for exactly realizing the real world in a virtual world, and on an underground utility tunnel digital twin that provides a service for disaster safety management. In this study, the methodology for implementing an underground utility tunnel, which is susceptible to disasters and accidents, is systematically defined using digital twin technology. Additionally, the study proposes a process for creating a model of the tunnel through this method.

## 2. Literature Review

The term "digital twin" has only been recently used explicitly in the engineering field, but the concept of a digital twin has long been applied in different professional fields. A well-known case of applying a digital twin is that of NASA's Apollo 13 mission in 1970, where the real-time malfunction data of the actual spaceship used for the moon exploration mission were input from the spaceship near the moon to the simulator on the ground through communications, and then the causes of failure and the restoration method of normal operation were investigated via a virtual experiment using the simulator [13,14]. Michael Grieves, a scholar of product lifecycle management, is known as the person who coined the term "digital twin" [15]. Currently, a digital twin is defined as a replica of physical assets, processes, and systems in which operational data, topography, space, shape information and movement, and operation rules of a replica and a target system are digitalized and saved on a computer [16,17].

Table 1 summarizes the cases in which a digital twin has been applied in the construction industry. The cases are largely classified into two fields, where the first case concerns the fields in which a digital twin is applied and the second case concerns the implementation process of a digital twin.

**Table 1.** Previous research cases on digital twins in the construction industry.

| REF | Year | Field | | | Application | | |
|---|---|---|---|---|---|---|---|
| | | Infra | Arch | City | Data Acquisition | Modeling | Service |
| [18] | 2021 | - | - | ● | QR codes, radio-frequency identification (RFID), IPC, sensors | 3D geometric model | Simulation Optimization |
| [19] | 2020 | ● | - | ● | GIS, sensing images, sensors, building management system (BMS) data | BIM | Monitoring, controlling the physical city |
| [20] | 2021 | ● | - | ● | Sensors, QR codes, space management system | BIM | Anomaly detection in operation and maintenance Blockchain |
| [21] | 2021 | ● | - | - | Geospatial datasets, demographic and climate conditions | BIM | Monitoring |
| [22] | 2021 | ● | - | - | Thermal imaging camera, sensors, AI | CIM, BIM, thermography map | Energy planning, simulation |
| [23] | 2019 | - | ● | ● | Sensors, financial compensation | 3D geometric model | Architecture for a cyber-physical system |
| [24] | 2019 | ● | - | - | Climate conditions, drawing data, geology data | BIM | Greenhouse gas emission, time schedule, cost data |
| [25] | 2013 | ● | - | | Magnitude of electrical current | 3D geometric model | Structural stability |
| [10] | 2020 | - | ● | - | LiDAR | 3D geometric model | City digital twin model |

●: Applied/–: N/A.

## 2.1. Digital Twin Field

Digital twins have been applied in various fields. In recent years, the use of digital twins has quickly expanded beyond the manufacturing industry to all industries including urban infrastructure, transportation, national defense, and aviation. Major countries are promoting the application of a digital twin as the core technology in relevant areas for securing future national competitiveness; some examples include the Virtual Singapore Project in Singapore and the National Digital Twin Program of the United Kingdom [26,27]. In the construction industry, digital twins have been applied to cities, buildings, and infrastructure. Research is being actively conducted on a digital twin of urban infrastructure, which has the nature of publicness, as well as in the construction industry [28].

Appendix A shows the revisions of the correlations in the construction industry that have been applied in previous studies. An underground utility tunnel has been added to the building/infrastructure level to highlight its importance. Previously, certain cases implemented underground facilities as a digital twin. Some studies proposed decision-making from the operation management perspective of tunnels [29], or examined design conditions related to the safety of tunnels [30]. Additionally, one study introduced geological analysis,

safety monitoring, and information on underground spaces, but there is a lack of research introducing the overall system involved in the process of building, linking, and servicing underground facilities and underground utility tunnels into a digital twin [31].

### 2.2. Digital Twin Application

#### 2.2.1. Data

Various types of data must be acquired in the process of initiating a digital twin. Data can be largely divided into three types: sociodemographic information, facility or geographic information, and sensor information. Sociodemographic information consists of population, demographic, and economic data as well as information on the floating population and management of facilities subject to the use of a digital twin [21]. Facility and geographic information, which is used in the modeling of a digital twin, is an essential element in the process whereby actual facilities are converged to virtual facilities [24]. Lastly, information on actual facilities and related data are collected through diverse sensors such as LiDAR [10].

#### 2.2.2. Modeling for Digital Twins

A modeling technique based on a 3D geometric model is required in the process of building a digital twin [18,23]. Various kinds of BIM-related software have been provided, and modeling is thus performed using BIM. In particular, the demand for 3D building models is increasing around the world, and the application fields of 3D building models are also becoming more diverse; thus, a higher significance is placed on a platform that can provide services for spatial information on a city model unit using 3D building models and topographic information. For providing a variety of services through a 3D spatial information platform, technology is needed for quickly visualizing large-scale data in a city model or large model unit [22].

#### 2.2.3. Services for Digital Twins

There is a need for a platform executing services based on data acquisition and modeling, and such a platform can use the platform of platforms (PoPs) structure, which employs interoperation. The PoPs structure enables the development and operation of all digital twins fitting the purpose of a digital twin, thus being efficient from the perspectives of reliability and economics (development period and cost) [20].

Table 2 presents the digital twin services provided in the construction industry, which are distinguished into general situations and disaster and emergency situations. General services include maintenance efficiency improvement, process optimization, cause analysis of system operation, and multi-party decision-making [19,25]. Disaster and emergency situations are classified as predictive maintenance and proactive control, in which proactive control is related to more sudden changes in states and emergency responses.

**Table 2.** Distinction between service types using a digital twin.

| Category | Major Services |
|---|---|
| General situations | Maintenance efficiency improvement |
| | Process optimization |
| | Cause analysis |
| | Multi-party decision-making |
| Disaster and abnormal situations | Predictive maintenance |
| | Proactive control |

Research is being conducted on how to respond to disasters at common construction sites, such as how to assess the risk of construction sites based on BIM [32], how to prevent geothermal disasters in underground spaces [33], and how to automatically extract risk

factors using deep learning [34]. In particular, safety accidents in an underground utility tunnel can escalate to hybrid disasters since a disaster happening in one lifeline can spread to other lifelines due to the dependence between facilities, thereby halting all city functions and hampering the everyday life of citizens, which eventually causes social confusion and foundation collapse; ultimately, infrastructure destruction, financial damage, and causalities can occur as a result [35].

## 3. Methods

The focus of this study is to propose a methodology for implementing a digital twin of an underground utility tunnel and to verify the proposed methodology through actual implementation. Figure 1 shows the overall methodology proposed in this study. First, the target space for research and demonstration is selected among real underground utility tunnels. Furthermore, three layers of data acquisition, digital modeling, and service are set for applying a digital twin of an underground utility tunnel.

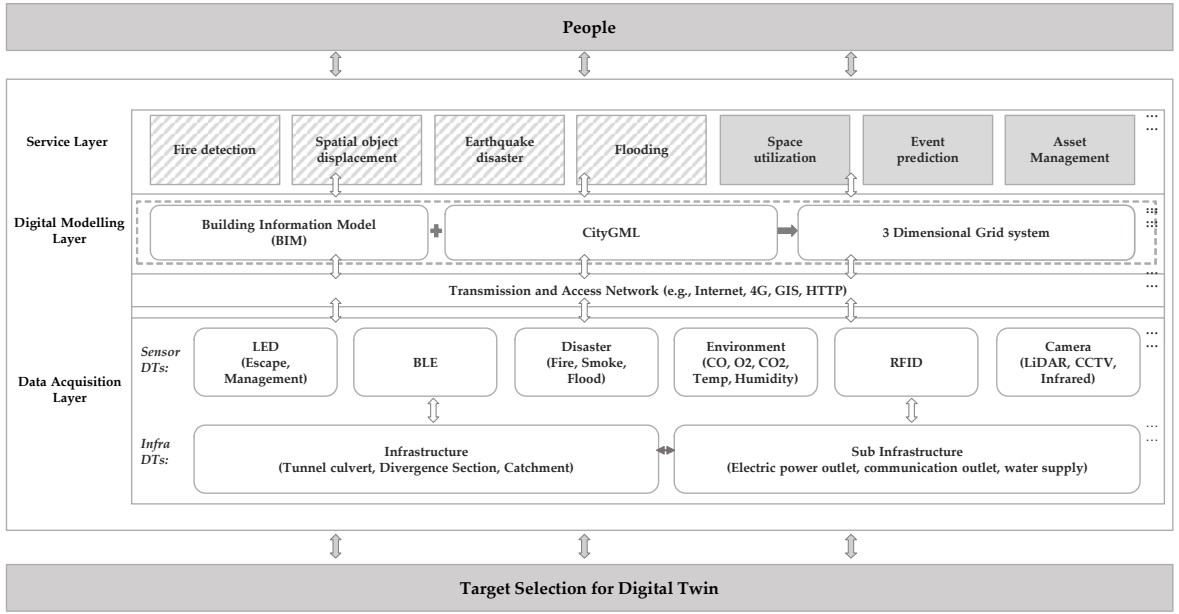

**Figure 1.** Methodology for constructing a digital twin in an underground utility tunnel.

### 3.1. Status of the Selected Underground Utility Tunnel

The Ochang utility tunnel has a length of 2426 m (3 sections, total length: 5586 m), of which the demonstration section has a length of 1210 m. Since the Ochang utility tunnel is situated in a harsh environment, 750 m of the power section where fire can break out, 60 m of the intense condensation section, and 400 m of the communications/water supply section were designated as the demonstration service sections for the entire section; the specific locations are shown in Figure 2.

To support the overall research process, BIM-based 3D modeling was performed based on the as-built drawings of the Ochang utility tunnel at the time of completion for the entire section (total of 5586 m including 3 sections), while LOD 4-based 3D modeling through a laser scan was performed only for the sections (1210 m) designated as the demonstration section.

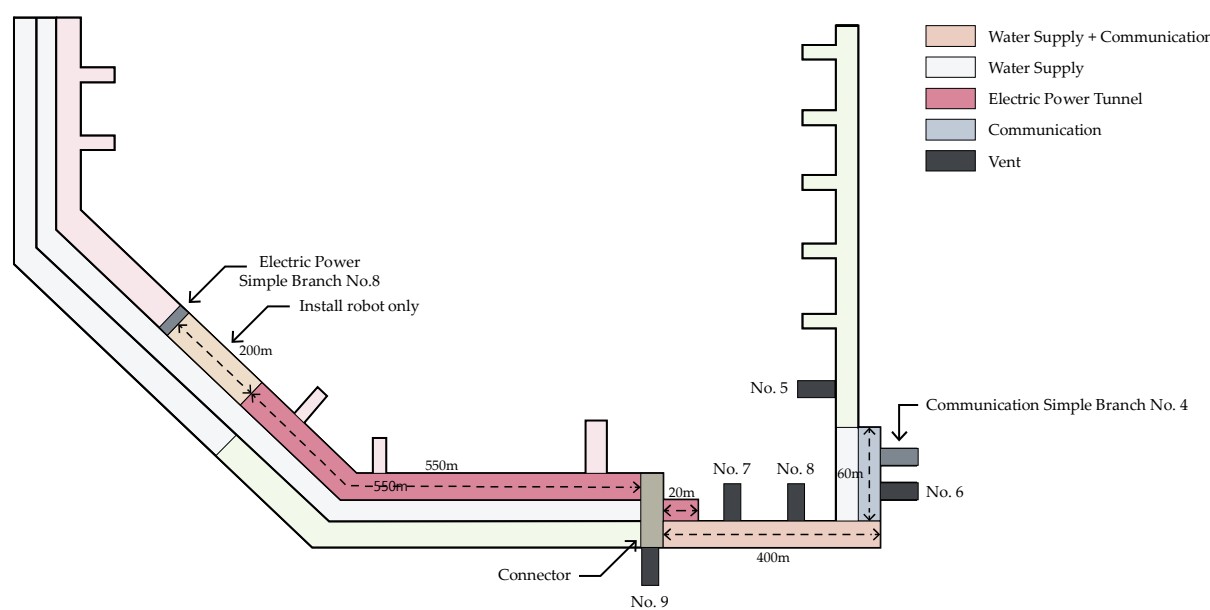

**Figure 2.** Floor plan of the target underground utility tunnel.

### 3.2. Key Methodologies for the UTU Digital Twin

Three layers were distinguished using the relevant methodologies summarized in Table 3. In the data acquisition layer, facility data, sensor data, and general information were collected. Facility data can be acquired based on existing 2D or 3D drawings or through a scanning process using LiDAR technology. The key role of the modeling layer is to model the underground utility tunnel using BIM and GIS. Additionally, a DB architecture needs to be configured for storing and linking information and service measures. Lastly, the service layer has distinct functions for general situations and abnormal situations. In particular, abnormal situations of underground facilities include fire detection, spatial object displacement, earthquake disaster, and flooding. The alerts for incidents in a digital-twin-based simulation can be generated based on pre-defined rules or algorithms that are designed to detect abnormal situations. For example, in a digital-twin-based simulation of an underground utility tunnel, the simulation could be programmed to detect abnormal situations such as a sudden temperature rise or a rapid increase in water levels. Once such situations are detected, the simulation can trigger alerts or notifications to relevant parties, such as the facility managers or emergency response teams. As for the dataset used in the simulation, it can include various types of data such as sensor data, geological data, and historical data on past incidents. These data are used to create a virtual representation of the physical environment and to simulate different scenarios to test the performance of the digital-twin-based simulation system.

**Table 3.** Key methodologies for the UTU digital twin.

|  | Main Features | Key Methodologies |
|---|---|---|
| Data acquisition layer | Information collection: infrastructure DT | LiDAR scan<br>2D documents<br>3D documents<br>Geospatial datasets |
|  | Information collection: sensor DT | Sensor data and image<br>QR codes |
|  | Information collection: general | Climate datasets<br>Demographic information<br>Maintenance datasets |

**Table 3.** *Cont.*

|  | Main Features | Key Methodologies |
|---|---|---|
| Modeling layer | Infrastructure and sensor modeling | BIM: library creation and color code designation |
|  | Geospatial modeling | GIS |
|  | Information storage and linkage measures | Database and linkage |
| Service layer | Abnormal situation management | Fire detection<br>Spatial object displacement<br>Earthquake disaster<br>Flooding |
|  | General situation management | Space utilization<br>Event prediction<br>Asset management |

## 4. Results

According to the methodology for implementing a digital twin of an underground utility tunnel explained in Section 3, data acquisition, modeling, and service layers were conducted, and the results are summarized as follows.

### 4.1. Data Acquisition Layer

4.1.1. Collecting Pre-Established Information

Two methods were applied for the 3D modeling of the Ochang utility tunnel. First of all, BIM-based 3D modeling was performed using as-built drawings of the Ochang utility tunnel at the time of completion for the entire section. The as-built drawings of the Ochang utility tunnel were categorized into two items and consisted of 423 CAD files for structures and facilities. These materials were well preserved as as-built drawings from the time of completion of the Ochang utility tunnel in 2001.

Site-based 3D modeling was performed via laser scanning for the demonstration section; the details related to the procedure of data acquisition via LiDAR scanning are shown in Table 4. Drawings and LiDAR scans are needed during the LiDAR scanning process, in which designating and mapping reference points of surveying are important. Lastly, point cloud matching is required in which 3D scan data are connected according to the site and for each station.

**Table 4.** Data acquisition procedure via LiDAR scanning and major details.

| Work Procedure | Major Details |
|---|---|
| Drawing printing | Check the work location on a numerical map<br>Prepare for a site inspection and survey |
| Site 3D scan | Site inspection and survey of facilities<br>Site inspection facility properties<br>Site exploration and survey before project launch |
| Select a survey position on the drawing | Select a survey position in CAD program according to the site inspection survey results |
| Survey ground-level reference point | Survey a precise reference point on the ground connected to the utility tunnel |
| Survey underground reference point | Link with the precise reference point surveyed on the ground |
| Reference point data mapping | Mapping between 3D scan data and reference point survey data |
| Point cloud matching | Connect 3D scan data to each station according to the site |

There are advantages and disadvantages of using as-built-drawing-based 3D modeling and LiDAR-based 3D modeling:

The advantages of 2D-based 3D modeling are as follows:

- Accessibility: 2D-based modeling is often easier and more accessible than LiDAR-based modeling as it can be created using basic software and traditional drawin tools.
- Cost-effective: 2D-based modeling can be less expensive than LiDAR-based modeling as it does not require specialized equipment such as LiDAR scanners.
- Simple: 2D-based modeling is simpler and easier to learn than LiDAR-based modeling, making it accessible to a wider range of users.

The advantages of LiDAR-based 3D modeling are as follows:

- High accuracy: LiDAR-based modeling can provide highly accurate 3D models of the underground utility tunnel, capturing details such as the precise location and geometry of objects and structures.
- High level of detail: LiDAR-based modeling can provide a high level of detail, including the position, size, and orientation of objects, which is important for simulating the operation and maintenance of the tunnel.

### 4.1.2. Site Information Collection

Site information of an underground utility tunnel can be collected through a variety of methods as shown in Table 5 below. Given the diverse nature of an underground utility tunnel, sensors can be attached based on their location and functionality characteristics, rather than their performance. The sensors can be classified into two categories—those inside and outside the underground utility tunnel—and further categorized based on whether they are fixedly installed, installed on moving robots, carried by people, or need to be connected to external systems.

**Table 5.** Overview of site information collection for underground utility tunnels.

| Monitoring Location | Monitoring Means and Method | Collected Data |
|---|---|---|
| Inside the underground utility tunnel | Fixed sensor: multisensors and cameras are repeatedly arranged at certain intervals in tens of km along the utility tunnel, and information is collected in real time | Temperature, humidity, oxygen, carbon dioxide, carbon monoxide, smoke, flame, vibration sensor, low-luminance image, thermal image |
| | Fixed sensor: sensor is arranged in facilities with a high risk | Vibration sensor of a water supply line, partial discharge sensor of a power line, water level sensor of a collection well, acoustic sensor |
| | Movable robot: robot moves on the ceiling rate over tens of km along the utility tunnel to collect information in real time | Temperature, humidity, oxygen, carbon dioxide, carbon monoxide, hydrogen sulfide, nitrogen dioxide, low-luminance image, thermal image, LiDAR sensor |
| | Manned patrol: patrol moves on foot and visually observes tens of km along the utility tunnel | Abnormalities in structures, firefighting facilities, power facilities, and communications facilities |
| Outside the underground utility tunnel | Fixed sensor: sensors and cameras are arranged at the entrance/exit of the management office and underground utility tunnel to collect information | Access security sensor, detection sensor of safety device |
| | Connection with external systems: regional and critical situation information is collected via the Meteorological Administration | Weather information such as temperature, humidity, precipitation, wind direction, wind speed, earthquake of a certain region, construction information, and terrorism risk information of the nearby areas |

For fixed sensors, multisensors and cameras are repeatedly arranged at certain intervals to collect information in real time. There are three major types of sensors used inside an underground utility tunnel: environmental ($CO_2$, $O_2$, temperature and humidity, smoke, and flame) and flooding sensors (ultrasonic sensors), location sensors (BLE sensors), access

sensors (RFID sensors), vibration sensors (gyro sensors), image sensors (low-luminance CCTV, thermal imaging cameras, and LiDAR sensors), and acoustic sensors.

Movable robots move along the ceiling rail to collect information in real time while the same tools as fixed sensors are used. Furthermore, various kinds of information are collected via manned patrol, which is a traditional method, and via connection with external (weather data) systems.

### 4.2. Modeling Layer

#### 4.2.1. BIM Modeling

The 3D modeling of the structure was performed for the entire section of the Ochang utility tunnel based on as-built drawings and design drawings from the Revit program, as shown in Figure 3. The procedure and results of 3D modeling are as follows. After modeling the structure, the facilities inside the underground utility tunnel were modeled. The major facilities were all modeled, including water supply pipes, power pipes, and communications pipes, as well as auxiliary facilities for repairing or managing the major facilities and other individual facilities such as fire extinguishers and CCTV for maintenance.

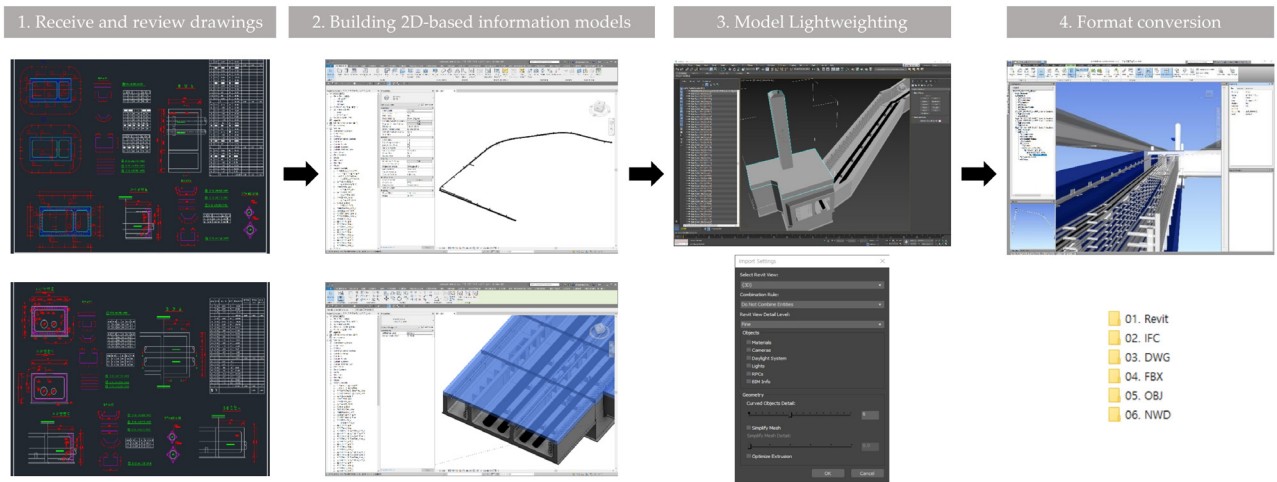

**Figure 3.** BIM model building procedure based on design drawings.

The as-built drawing of the demonstration section must reflect the final revision of the structure, unlike the design drawing. In most cases in the past, however, the final state at the time of completion was frequently not reflected in the design drawing. For the same reason, the Ochang underground utility tunnel also had certain sections showing differences between the as-built drawing and the site scanning results; the data differences were confirmed by overlapping the two types of data. The sections with differences between the site survey results and the CAD-based BIM model were remodeled based on the site survey results, thus modeling the structure to match the site. Figure 4 shows the process of building a BIM model based on laser scanning.

The collected drawing materials are in a form that can be generally used for work. However, additional property information is needed for facility management through 3D modeling. Therefore, property items to be modeled for the libraries being configured are shown in Figure 5.

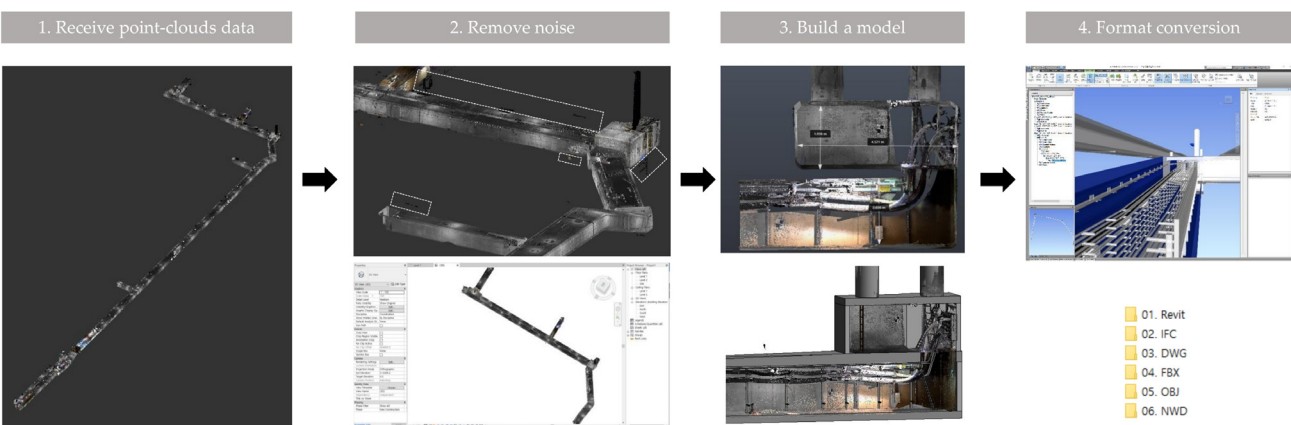

**Figure 4.** BIM model building procedure based on a laser scan.

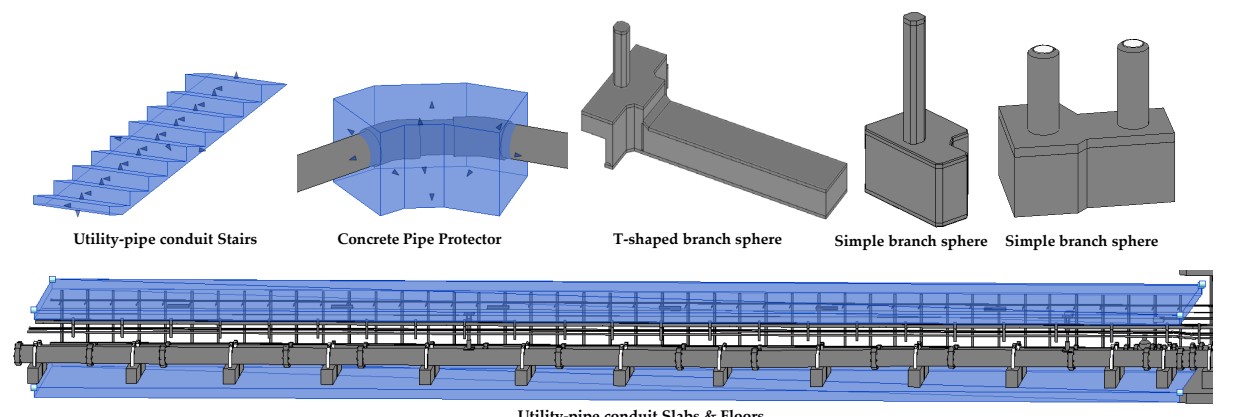

Structure library

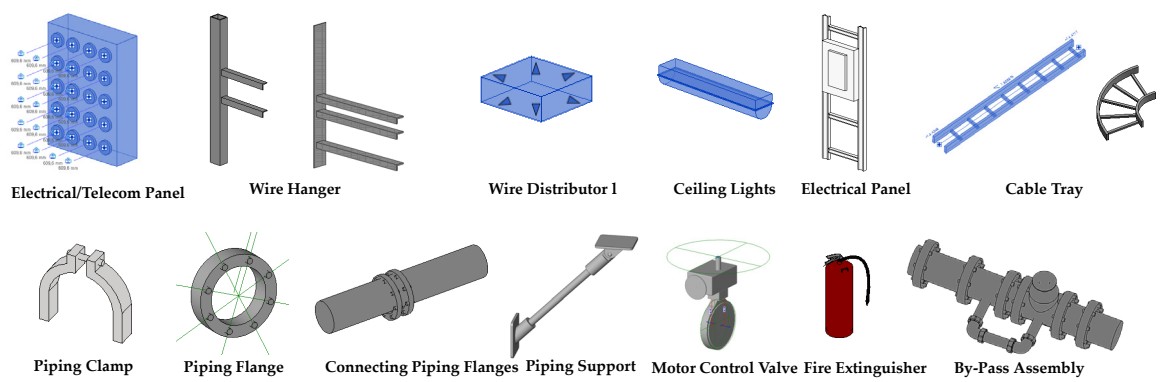

Facility library (1/2)

**Figure 5.** *Cont.*

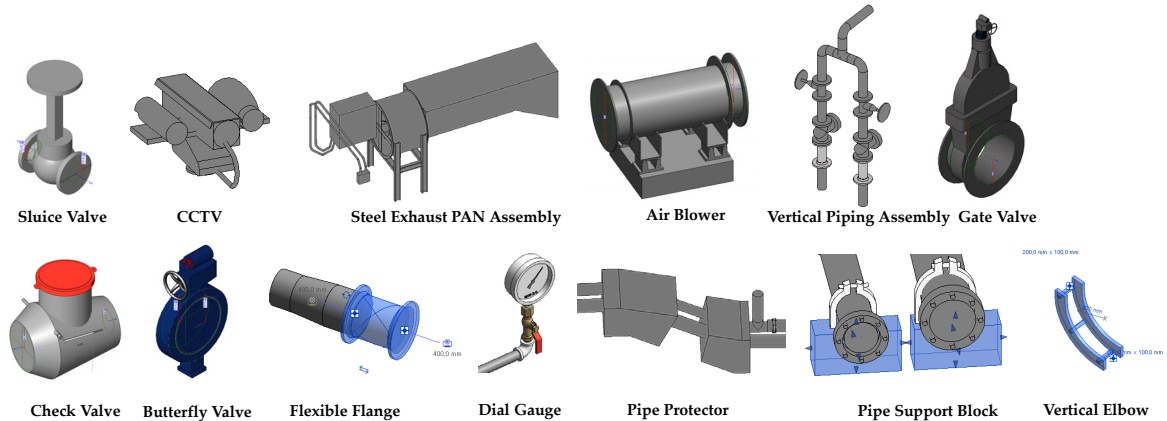

Facility library (2/2)

**Figure 5.** Underground utility tunnel BIM library.

Users can quickly identify each object when the visibility of a 3D model is heightened by inputting standardized color codes according to the spatial object unit classification within the 3D model of an underground utility tunnel based on the 3D scanning of the demonstration section. The 3D model was further classified, as shown in Figure 6.

The color codes were defined according to the regulations related to the underground facility digital mapping project; the BIM data standard was established primarily based on the color codes devised by machine type, firefighting work type, and electrical/communication work type. The standard method for devising facility BIM data in this project is shown in the following table. The facility BIM data standard consists of the expression standard as well as property information definitions by machine, electric, and member type; the expression standard is subdivided into machine and electric. The requirements of the property information of underground utility tunnel facilities are defined by machine, electric, and member type.

The drawing notation of each process was defined in addition to the color codes of an underground utility tunnel; 3D model objects were identified by applying RGB color codes according to the characteristics of the work type.

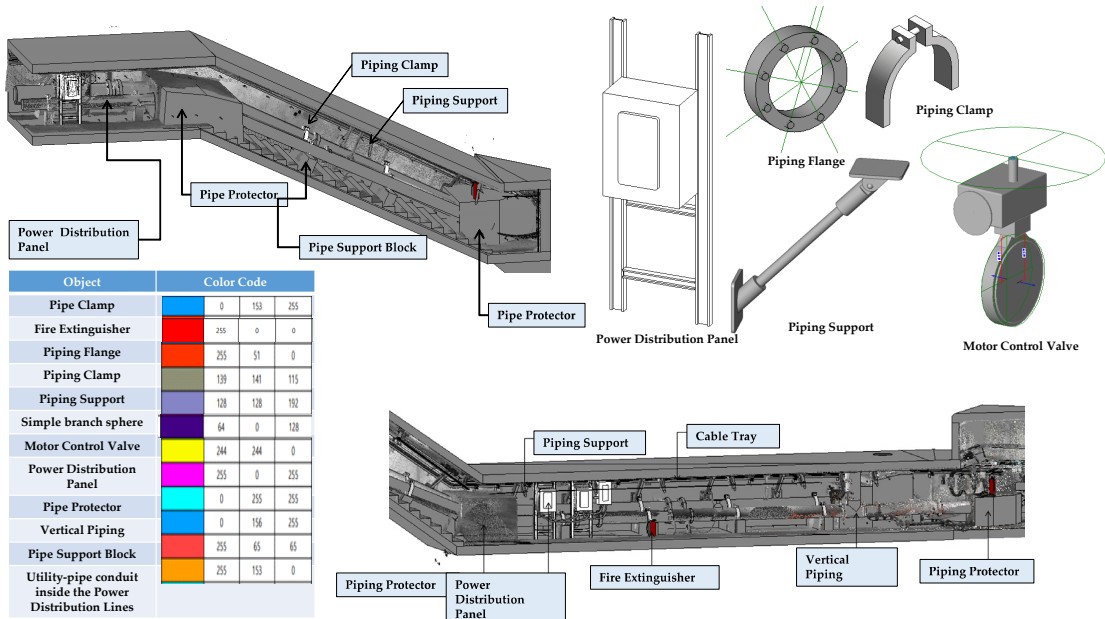

| Object | Color Code | | |
|---|---|---|---|
| Pipe Clamp | 0 | 153 | 255 |
| Fire Extinguisher | 255 | 0 | 0 |
| Piping Flange | 255 | 51 | 0 |
| Piping Clamp | 139 | 141 | 115 |
| Piping Support | 128 | 128 | 192 |
| Simple branch sphere | 64 | 0 | 128 |
| Motor Control Valve | 244 | 244 | 0 |
| Power Distribution Panel | 255 | 0 | 255 |
| Pipe Protector | 0 | 255 | 255 |
| Vertical Piping | 0 | 156 | 255 |
| Pipe Support Block | 255 | 65 | 65 |
| Utility-pipe conduit inside the Power Distribution Lines | 255 | 153 | 0 |

**Figure 6.** Color code input.

4.2.2. Building and Operating System of a Digital Twin Space

To secure the interoperability of the 3D model data in this study, the IFC standard, which is the BIM standard format, was applied; to secure the interoperability of the digital twin of an underground utility tunnel, this study proposes the OGC standard (OGC 3D Tiles Specification)-based 3D spatial data model (DTS-DM) for linking BIM data and GIS data. Furthermore, a real-time digital twin space service is supported based on a 3D spatial data model according to the OGC 3D tiles standard, and an open platform that can utilize various application services was developed.

The digital twin spatial data model DTS-DM proposed in this study consists of a database schema that can store BIM data for effective inquiry and search in large-scale BIM data and GIS data, as well as prompt screen displays for the interoperability of BIM and GIS data. The DTS-DM model, which was built based on the OGC 3D Tiles Specification standard, is a BIM/GIS spatial data model that can express exterior representation objects, the LOD concept of CityGML (GIS), and information on interior building elements of IFC. Figure 7 shows the concept of DTS-DM.

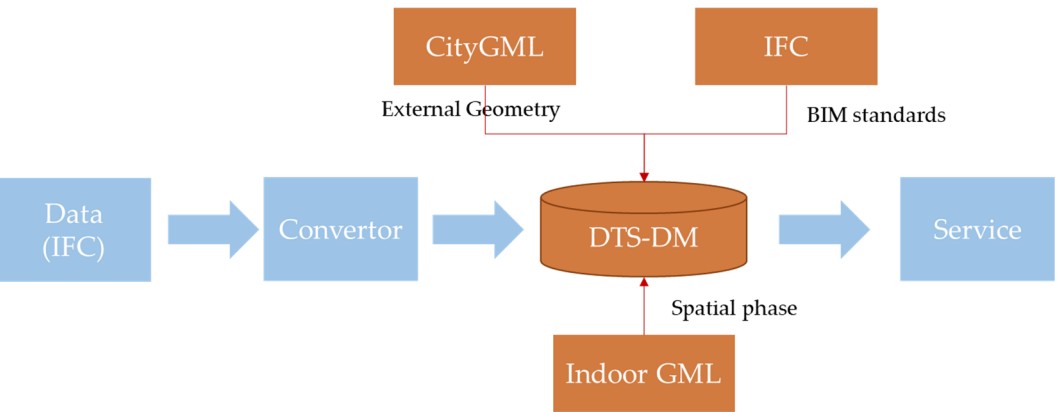

**Figure 7.** Underground utility tunnel DTS-DM conceptual model.

DTS-DM defines and expresses an underground utility tunnel (UCA) region as a space model. Each space model consists of the exterior model of such elements as ceiling and wall, and the interior model of such elements as stairs, doors, and internal structures. Furthermore, linear facilities such as water supply lines, communications lines, and power lines that are crucial facilities of an underground utility tunnel are defined as lifeline models, while sensor devices such as CCTV are defined as sensor models.

To build a digital twin of an underground utility tunnel, a DTS-DM-based digital twin space is built and a platform system is established for real-time services; the overall flow of the system service is shown in Figure 8. The platform system receives the original data of the underground utility tunnel model in the IFC form and then converts it to the DTS-DM digital twin spatial data model for providing services to users. Service data converted by DTS-DM largely consist of OGC 3D tiles, which have the hierarchical file structure per LOD, spatial shape for renewal, and database for storing properties. Users can perform a high-speed search by using the metadata (index structure) of a 3D tile structure and can then use the services of a digital twin space.

*4.3. Service Layer*

4.3.1. Service Items of an Underground Utility Tunnel

Underground utility tunnel service technology using a digital twin involves analyzing problems that may occur in an underground utility tunnel through big data, AI, modeling, and simulation, deducing optimization measures, and applying them in the field. Underground utility tunnel management technology using a digital twin consists of the underground utility tunnel management service and response service for abnormal situa-

tions. The underground utility tunnel management service is intended to provide manned and unmanned surveillance, diagnoses, and risk inference services through a digital twin. The underground utility tunnel abnormal situation response service distinguishes disaster stages into a warning, initial response, and active response, and consists of a standard operation process (SOP) for responding to abnormal situations and a service for effectively responding to abnormal situations by comprehensively analyzing site situation information. Therefore, the underground utility tunnel abnormal situation management platform based on a digital twin aims to provide integrated services from the disaster prediction of fire, spatial object modification, and earthquake to response and survey analysis; thus, the services are provided in order of data collection, prediction, prevention, response, and survey analysis.

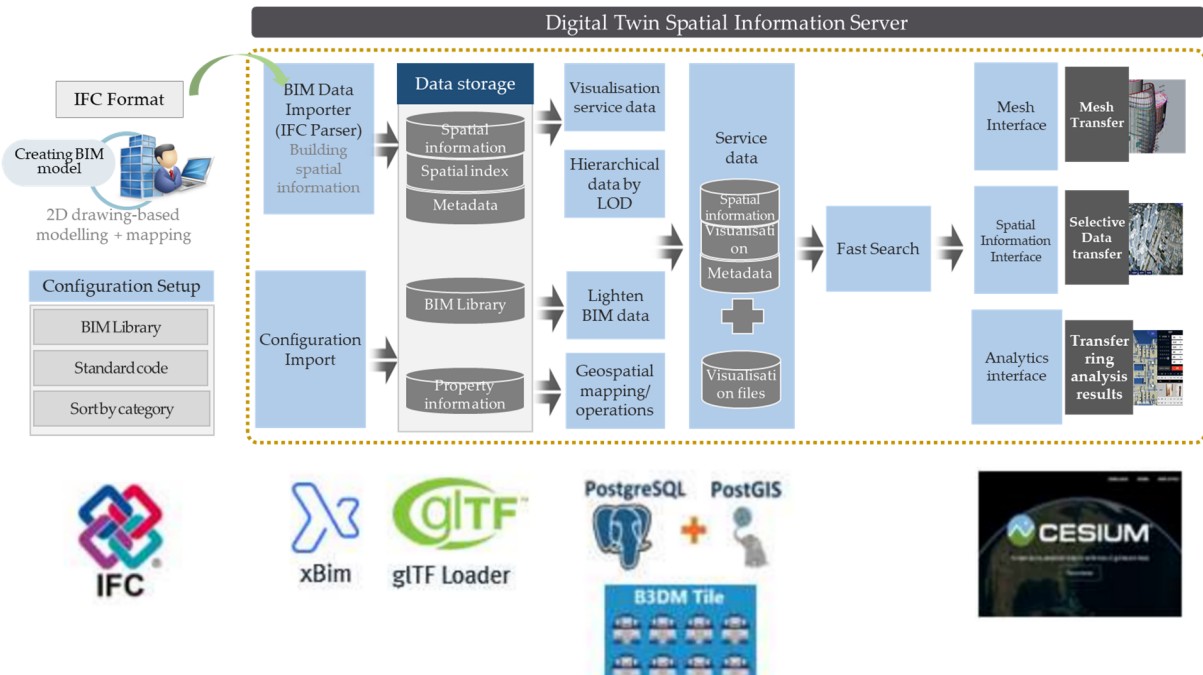

**Figure 8.** Service flow of a digital twin space.

### 4.3.2. Service Functional Requirements of an Underground Utility Tunnel

As shown in Table 6, the required functions of an underground utility tunnel digital twin system were deduced according to the scenario when using an underground utility tunnel digital twin system based on the scenario of the situation room and site workers. For system classification, digital twin functions can be distinguished into DB management and data transmission, underground utility tunnel management, safety management, abnormal signal transmission, and simulation. It was classified based on the minimum required functions of the system according to the requirements of underground utility tunnel managers and scenarios; the minimum required functions can be applied when building an underground utility tunnel digital twin system.

**Table 6.** Functional requirements of an underground utility tunnel digital twin.

| Category | Description |
| --- | --- |
| | Collect and store databases |
| Database management | Convert and transmit databases |
| | Statistical analysis of databases |

**Table 6.** *Cont.*

| Category | Description |
| --- | --- |
| Facility management | Status inquiry of management and accommodation facilities |
| | Inspection of management and accommodation facilities |
| | State information display of management and accommodation facilities |
| | Search firefighting facility control data |
| | CCTV control |
| | Register field situations |
| | Inquiry of workers' locations |
| Safety management | Check for abnormalities |
| | Request on-site support |
| Abnormal signal management | Check for abnormal situations |
| | Propagate situations |
| Simulation management | Guide evacuation routes |
| | Suppress disaster spread |
| | Predict damage spread |
| | Visualization of analysis results |

The underground utility tunnel manager, accommodation facility manager, and disaster response system manager, which are the recipients of the abnormal situation message and digital twin model, should constantly identify the operational status of an underground tunnel, abnormal situations, and disasters based on abnormal situation message and decision-making support information provided in stages, and immediately execute response measures such as promptly giving commands and reporting in cases of emergency. Specifically, event messages of abnormal situations and the digital twin model must be presented in an easy-to-comprehend manner and be readily controllable by information recipients. An underground utility tunnel is an underground facility having a length of tens of km in general, and therefore it is extremely challenging to respond to disasters. Therefore, an underground utility tunnel digital twin platform should realistically reflect 3D spatial information to facilitate exploring an underground utility tunnel in three dimensions and support AR-based mobile patrol and inspection, as well as the disaster simulation of fires, flooding, and earthquakes reflecting the actual structure and sizes of a complex underground utility tunnel, and should report the operational status of accommodation, electric, and communications facilities at the time of disaster.

## 5. Conclusions

This study systematically defined the methodology of implementing an underground utility tunnel, which is vulnerable to disasters and accidents, through a digital twin, and proposed the process for generating such a model. Since a digital twin has been increasingly applied for maintenance and asset management in the construction industry. This study aimed to sequentially implement the digital twin for underground facilities. Therefore, the major layers needed for configuring an underground utility tunnel digital twin are data acquisition, modeling, and service layers, which have been analyzed according to each process as follows.

- Data acquisition: The key role of the data acquisition layer is collecting facility data, sensor data, and general information. Facility data can be acquired based on existing 2D or 3D drawings or through a scanning process using LiDAR technology.
- Modeling: Modeling is proceeded based on data building. The key role of the modeling layer is to model the underground utility tunnel by using BIM (infrastructure

and sensor data) and GIS (spatial information). Additionally, modeling was efficiently performed by configuring a DB architecture for storing and linking relevant data and for providing services.

- Service: Lastly, underground utility tunnel management technology using a digital twin consists of an underground utility tunnel management service and a response service for abnormal situations. An underground utility tunnel management service provides manned and unmanned surveillance, diagnoses, and risk inference services through a digital twin. In particular, abnormal situations in underground facilities include fire detection, spatial object displacement, earthquake disaster, and flooding.

This study is significant in that the findings and proposed methodology can be utilized in the future when implementing a digital twin of underground facilities and underground utility tunnels. Disasters or abnormal situations in an underground utility tunnel can lead to serious casualties and property damage since key elements of a city such as power, communications, water supply, and heating facilities are collectively accommodated in a certain underground space. As a further step, the digital twin model proposed in this study needs to be validated in terms of its ability to accurately represent the actual site and reproduce real-world situations. In this process, it is necessary to validate and compare the amount of data actually built into the database, the accuracy of the 3D model, and its ability to accurately depict abnormal situations.

In particular, studies must be additionally performed for abnormal situations in which research must focus on distance and object recognition and the displacement detection of facilities and structures based on a digital twin.

**Author Contributions:** Conceptualization, J.L. and C.H.; methodology, J.L. and C.H.; investigation, J.L., C.H. and Y.L.; data curation, J.L.; writing—original draft preparation, J.L. and Y.L.; writing—review and editing, J.L. All authors have read and agreed to the published version of the manuscript.

**Funding:** This research was supported by an Institute of Information and Communications Technology Planning and Evaluation (IITP) grant funded by the Korean government (MSIT, MOIS, MOLIT, and MOTIE) (No. 2020-0-00061, Development of integrated platform technology for fire and disaster management in underground utility tunnel based on digital twin).

**Institutional Review Board Statement:** Not applicable.

**Informed Consent Statement:** Not applicable.

**Data Availability Statement:** The datasets used and analyzed during the current study are available from the corresponding author upon reasonable request.

**Conflicts of Interest:** The authors declare no conflict of interest.

## Appendix A

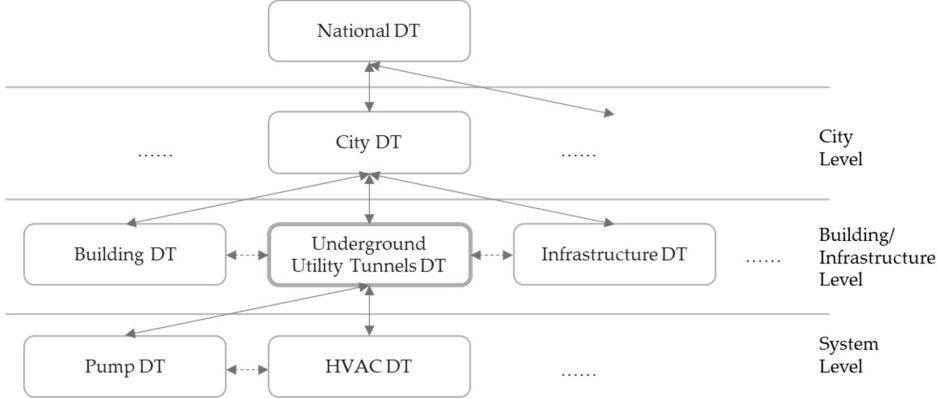

**Figure A1.** Connections, hierarchy, and level of digital twins [19].

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
