# Peer review of "Development of Geospatial Data Acquisition, Modeling, and Service Technology for Digital Twin Implementation of Underground Utility Tunnel"

_applsci, doi:10.3390/app13074343_

Round 1

Reviewer 1 Report

The authors have presented a method of detecting, monitoring and responding to distress and disasters in an underground utility tunnel by using Building Information Modeling and Digital Twin. The proposed method can add to the existing body of knowledge of automated visualization techniques in the construction industry. The following comments need to be addressed:

1. In the Introduction section, along  identification of the research gap, the authors are recommended to write a sentence or two describing the work performed in this paper. The reader will otherwise be left guessing about the specifics of the work performed until Methods.

2. In the Literature section, some more papers can be referenced related to the use of AR and VR and other smart technologies in the field of construction with an emphasis on disaster preparedness.

3. The service layer needs to be better defined. How were the alerts for incidents simulated? What kind of dataset was used?

4. How can the proposed model (or some parts of it) be validated? Is it possible to show validation under the scope of the current manuscript? If not, please specify how further studies by others can validate the method and/or model.

Author Response

Dear Reviewer,

We are grateful to referees for the valuable comments that helped us improve the clarity of our manuscript. We have addressed the reviewers’ requests and hope the paper is now ready for publication.

Title of the paper: Development of Geospatial Data Acquisition, Modeling, and Service Technology for Digital Twin Implementation of Underground Utility Tunnel

Reviewer 2 Report

This paper addresses the topic of a digital twin system development for the underground utility tunnel, which contains data acquisition, modeling, and service layers. This paper provides a reference value for the implementation of digital twin in underground facilities in the future. The topic of the paper is interesting, but several modifications are required.

1. Section 2 has less literature review on the management and disaster response of underground utility tunnel systems. Some more references that could be included and explain the advantages of using the methodology of this paper.

2. Figure 1 is not obvious enough in reflecting the importance of digital twin in this field.

3.Section 4.1.1, the two 3D modeling methods should be explained clearly of the advantages and disadvantages and what scenarios are applicable? The point cloud matching algorithm mentioned in section 4.2, how to ensure the accuracy of matching?

4. Section 4.1.2, the description of sensor types is not clear, what are the specific categories?

5. Moreover, some abbreviations must first be introduced in the paper with their full names. Some of the figures require improvements as they are not legible for the reader (e.g. Figure 9, etc.)

6. Section 5 should also list the main limitations of the method in this paper. The conclusions should be more transparent in this work.

7. Please ask a native English speaker or a professional language service to proof read and improve the use of English language in your paper.

Author Response

(The authors gave the same response as above.)

Round 2

Reviewer 2 Report

The Submission has been greatly improved and is worthy of publication.